# GRASP: Grouped Activation Shared Parameterization for Parameter-Efficient Fine-Tuning and Robust Inference of Transformers

## Abstract

Parameter-efficient fine-tuning (PEFT) provides a scalable alternative to full-model adaptation by updating only a small subset of parameters in large pre-trained models. We introduce **GRASP** — GRouped Activation Shared Parameterization — a lightweight PEFT framework that partitions the $D$-dimensional token representations of selected layers into $K \ll D$ groups and learns a shared scaling and shifting vector for each group. This grouped modulation reduces the number of trainable parameters significantly while preserving the ability of the model to learn task-specific features. Across GLUE (RoBERTa-base & RoBERTa-large) and E2E NLG (GPT-2 Medium), GRASP demonstrates competitive performance compared to established PEFT methods while achieving an order of magnitude reduction in trainable parameters compared to LoRA and BitFit. Analyzing the learned parameters, we observe that aggressive grouping (small $K$) transforms their layer-wise distribution from unimodal to multimodal, indicating that the method encodes distinct task-relevant structure under tight parameter budgets. Motivated by this, we further propose **StochGRASP**, which learns Gaussian distributions as perturbations to the pre-trained weights rather than deterministic values. This probabilistic parameterization along with a noise-aware loss function formulation enables modeling hardware-level variability in programmed weights and improves robustness under non-ideal inference conditions—an important requirement for deployment on edge-based emerging AI hardware. In preliminary experiments under simulated inference-time noise, StochGRASP consistently outperforms deterministic variants, demonstrating its suitability for energy-efficient and noise-prone hardware platforms.

## 1 Introduction

Pre-trained transformers (Kalyan et al., 2021; Vaswani et al., 2017) form the backbone of modern foundation models (Zhou et al., 2025), but their large number of trainable parameters presents significant challenges for domain specific adaptation (Guo & Yu, 2022)—particularly in low-resource scenarios (Hu et al., 2021). Moreover, on tasks with limited data, full fine-tuning often leads to overfitting, resulting in reduced generalization and lower inference-time accuracy (Liu et al., 2022).

To tackle these challenges, parameter-efficient fine-tuning (PEFT) (Xu et al., 2023) methods have emerged as effective strategies for adapting large models by updating only a small subset of parameters. This approach significantly reduces computational costs, storage requirements, and training time. Widely adopted PEFT techniques such as LoRA (Hu et al., 2021) and Adapters (Poth et al., 2023) achieve strong performance but introduce additional matrix multiplications during training and typically require more parameters than methods based on lightweight, element-wise shifting & scaling operations. Notable alternatives in this category include BitFit (Zaken et al., 2021), which fine-tunes only the bias terms, and $(IA)^3$ (Liu et al., 2022), which learns scaling vectors for selected layers. Although these methods achieve a favorable balance between efficiency and performance, they still incur a trainable parameter cost of the order of $O(n \times D)$, where $n$ is the number of layers and $D$ represents the hidden dimension (either the model or intermediate size), indicating potential for further compression.

In this paper, instead of learning separate linear scale & shift parameters for each of the $D$ components, we group them into $K$ clusters and assign shared parameters within each group. For chosen linear projection layers where we apply GRASP, we modulate the input activations to the layer using this shared set of parameters. This grouped modulation approach reduces the number of trainable parameters from $\mathcal{O}(n \times D)$ to $\mathcal{O}(n \times K)$, where $K \ll D$. Analogous to how LoRA demonstrates that fine-tuning can be achieved by learning low-rank ($r << D$) updates to weight matrices, our method learns only $K$ group-wise scaling and shifting parameters. While LoRA introduces approximately $\mathcal{O}(D \times r)$ trainable parameters per layer (where $r$ is the low rank), our approach requires only $\mathcal{O}(K)$ parameters per layer, offering an even more compact and efficient alternative for PEFT.

While the primary contribution of GRASP lies in reducing the computational overhead of PEFT by drastically lowering the number of trainable parameters on GPUs, a deeper analysis of the layer-wise parameter distributions reveals a consistent emergence of structured, multimodal patterns. Motivated by this observation, we reinterpret PEFT as the task of learning *underlying perturbation distributions* that can be sampled to adapt the frozen pre-trained weights to a new dataset.

Building on this perspective, we introduce **StochGRASP**, a PEFT strategy that learns Gaussian parameter distributions instead of fixed deterministic updates. This distributional formulation yields perturbations that are inherently more resilient to device/circuit-level variability—an important requirement for deployment on noisy or non-ideal hardware. For instance, emerging energy-efficient AI hardware accelerators based on novel non-volatile memories suffer from multiple sources of noise (device-to-device variations, cycle-to-cycle variations, drift effects) and mitigation of such non-idealities is an active area of research (Yousuf et al., 2025; Manna et al., 2024). To enable robust variation-immune inference on such edge AI hardware platforms, we explore a noise-aware fine-tuning objective that regularizes the learned standard deviations toward a target noise profile, encouraging the model to maintain stable performance across a range of hardware noise conditions. The primary contributions of this work are:

1. We introduce **GRASP** (GRouped Activation Shared Parameterization), a lightweight PEFT method that partitions the $D$-dimensional hidden representation into $K$ groups and learns only a shared scaling and shifting pair per group, significantly reducing the number of trainable parameters.

2. Through a layer-wise analysis of parameter distributions, we show that decreasing group size transforms the learned parameters from a unimodal to a multimodal structure, highlighting GRASP's ability to retain task-specific expressivity despite aggressive compression.

3. GRASP matches or exceeds prior element-wise and low-rank PEFT methods on masked language understanding (GLUE with RoBERTa-Base/Large) while training up to 75× fewer parameters than LoRA and 12–22× fewer than (IA)$^3$ and BitFit; on causal generation (E2E NLG with GPT-2 Medium) it attains a favorable accuracy-for-parameters trade-off, and in the low-parameter regime outperforms rank-1 LoRA, rank-1 Adapters, and RED at a fraction of their trainable parameters.

4. Motivated by the emergent multimodal structure of the learned parameters, we reinterpret PEFT as learning *distributions* of weight perturbations rather than deterministic updates, yielding StochGRASP. In preliminary experiments under simulated inference-time noise, StochGRASP exhibits substantially improved tolerance relative to a deterministic baseline, suggesting a promising direction for robust deployment on noise-prone edge and analog AI hardware.

## 2 Related Works

Parameter-efficient fine-tuning (PEFT) has emerged as a powerful strategy for adapting large pre-trained models while updating only a small fraction of parameters. Existing PEFT techniques can be broadly categorized into *layer-selective*, *module-based*, and *element-wise* approaches.

**Layer-selective methods** fine-tune only a subset of transformer layers to reduce training cost. For example, FFTTop2 (Hu et al., 2021) updates only the top two layers of the model while keeping the remaining parameters frozen.

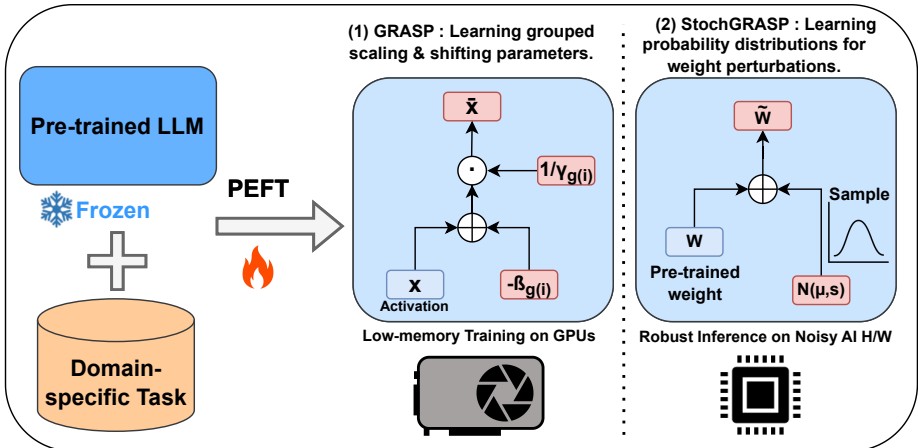

Figure 1: High-level overview of the proposed PEFT framework. (1) GRASP performs parameter-efficient fine-tuning by learning grouped scaling and shifting parameters over activations, enabling low-memory training with deterministic modulation. (2) StochGRASP extends this concept by learning Gaussian perturbation distributions over weight updates rather than fixed values, coupled with a noise-aware objective that yields significantly more robust inference under hardware-level noise than deterministic baselines.

**Module-based PEFT** introduces small trainable components into the network. Adapters (Poth et al., 2023) insert bottleneck layers inside transformer blocks and update only these modules during fine-tuning. LoRA (Hu et al., 2021) replaces full-rank updates with low-rank matrices, substantially reducing the number of trainable parameters while preserving representational flexibility. Other techniques, such as Prefix Tuning (Li & Liang, 2021), optimize continuous prefix vectors at the embedding layer rather than modifying internal weights.

**Element-wise modulation methods** operate directly on hidden activations and avoid introducing additional matrix multiplications. BitFit (Zaken et al., 2021) tunes only bias terms, whereas $(IA)^3$ (Liu et al., 2022) learns multiplicative scaling vectors applied to selected projections. RED (Wu et al., 2024) extends this family by learning both scaling and shifting parameters on transformer activations.

**Uncertainty-aware or stochastic modeling** within transformers is also an active area of research. BayesFormer (Sankararaman et al., 2022) and related variational dropout methods estimate predictive uncertainty, but their focus is on improving generalization rather than enabling robustness under device-level noise or analog hardware variability. While there have been works on developing variation-resilient models for such analog AI hardware platforms (Yousuf et al., 2025; Manna et al., 2024), the existing probabilistic models remain limited on small scale vision-datasets and simple convolutional networks. **Our work uniquely bridges PEFT with variation-aware robust inference, addressing a critical gap in existing approaches.**

## 3  Grouped Activation Shared Parameterization

GRASP aims to adapt pre-trained transformer models by freezing the majority of model weights and applying lightweight linear scaling and shifting operations to the hidden state activity of a sub-layer that we want to fine-tune. Unlike prior methods such as BitFit/$(IA)^3$, which learn individual shift/scale parameters across the entire hidden dimension, GRASP learns only $K$- pairs of parameters where $K \ll D$. As a result, all $D/K$ dimensions within the same group share the same modulation parameters (Fig. 2a), drastically reducing the number of trainable parameters while achieving competitive results.

Our proposed method introduces no additional matrix multiplications during training, resulting in lower computational overhead compared to methods such as LoRA and Adapters—while also significantly reducing

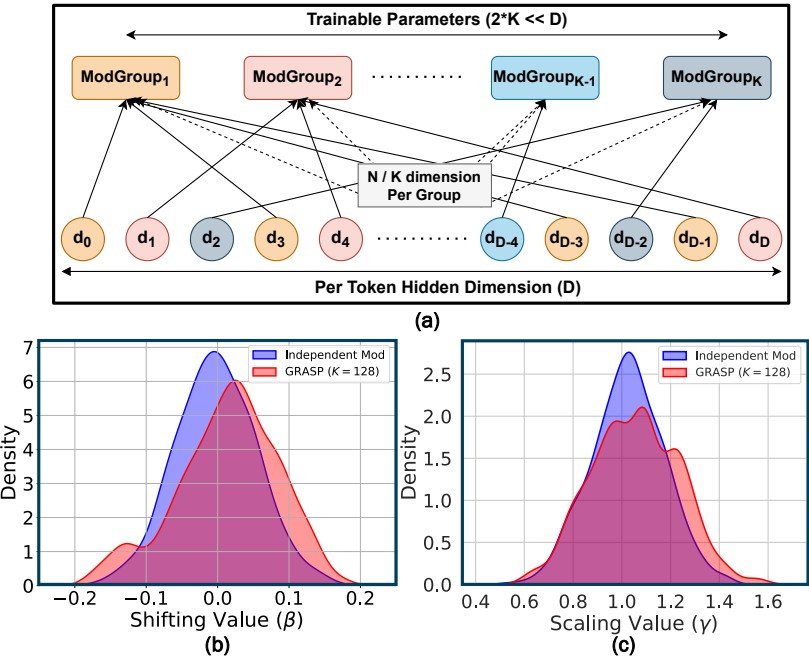

Figure 2: (a) Random grouping of per-token $D$-dimensional input activation to a layer into $K$ groups. GRASP learns scaling and shifting parameters per group rather than independently for each component. (b & c) Kernel Density Estimation (KDE) plot showing (b) the distribution of shifting parameter ($\beta$) and (c) the distribution of scaling parameter ($\gamma$) learned in a selected projection layer (Key) when parameters are learned independently versus using GRASP with $K = 128$. . Results are from GLUE CoLA dataset.

the number of trainable parameters. For each projection layer selected for fine-tuning, we modulate the input activation as follows:

$$
\begin{aligned}
\tilde{x}_i^\tau &= \frac{x_i^\tau - \beta_{g(i)}}{\gamma_{g(i)}} \quad \text{for } i = 1, 2, \ldots, D_{in}, \\
\tilde{Y} &= \tilde{X} W
\end{aligned}
\tag{1}
$$

where $x_i^\tau$ denotes input activation, i.e., the output of the previous layer, for the $i$-th dimension of token $\tau$, and $\tilde{x}_i^\tau$ represents the entire modulated input. $\tilde{X} \in \mathbb{R}^{L \times D_{in}}$ is the modulated input, $W \in \mathbb{R}^{D_{in} \times D_{out}}$ is the frozen layer weight and $\tilde{Y} \in \mathbb{R}^{L \times D_{out}}$ is the output, where $L$ is sequence length, $D_{in}$ is the activation dimension of the previous layer and $D_{out}$ is the activation dimension of the current layer. $g(i) \in \{1, 2, \ldots, K\}$ maps each dimension component in $D_{in}$ to one of $K$ groups, with $\gamma_{g(i)}$ and $\beta_{g(i)}$ denoting the learnable scaling and shifting parameters shared across all components within group $g(i)$.

## 3.1 Random Grouping

Instead of doing any computationally expensive pre-processing to learn the grouping function ($g(i)$), we propose an effective strategy to randomly group dimensions. The method used in this paper is described below,

$$
\begin{aligned}
\tilde{\gamma} &= \left( \gamma^{\oplus m} \right)_{1:D}, \quad \tilde{\beta} = \left( \beta^{\oplus m} \right)_{1:D} \\
\hat{\gamma} &= \tilde{\gamma}[\pi], \quad \hat{\beta} = \tilde{\beta}[\pi]
\end{aligned}
\tag{2}
$$

| Model | Method | %Param | QNLI | SST-2 | MNLI | CoLA | MRPC | STS-B | RTE | QQP | Avg. |
|---|---|---|---|---|---|---|---|---|---|---|---|
| | Full-FT (Zaken et al., 2022) | 100 | 92.3 | 94.2 | 86.7 | 61.1 | 92.5 | 90.6 | 77.4 | 91.9 | 85.8 |
| | Adapters (Hu et al., 2021) | 0.30 | 93.1 | 94.2 | 87.1 | 60.8 | 88.5 | 89.7 | 71.5 | 90.2 | 84.4 |
| | LoRA (Updated) (Hu et al., 2021) | 0.30 | 93.3 | 95.1 | 87.5 | 63.4 | 89.7 | 91.5 | 86.6 | 90.8 | 87.2 |
| | AdaLoRA (Zhang et al., 2023) | 0.30 | 93.0 | 94.8 | 87.6 | 63.2 | 88.8 | 91.1 | 79.9 | 90.1 | 85.9 |
| RoBERTa-Base | DoRA (Mao et al., 2024) | 0.30 | 93.0 | 95.0 | 87.5 | 64.9 | 89.7 | 91.3 | 79.2 | 90.6 | 86.4 |
| | BitFit (Zaken et al., 2022) | 0.09 | 91.3 | 93.7 | 85.0 | 61.8 | 92.0 | 90.8 | 77.8 | 87.3 | 84.9 |
| | $(IA)^3$ (Liu et al., 2022) | 0.05 | 91.1 | 93.4 | 85.4 | 57.8 | 86.4 | 88.5 | 73.5 | 88.5 | 83.1 |
| | RED (Wu et al., 2024) | 0.02 | 90.7 | 93.9 | 83.9 | 61.0 | 89.2 | 90.4 | 78.0 | 87.2 | 84.3 |
| | **GRASP** ($K$=128) | **0.015** | **92.0** | **94.3** | **85.2** | **64.2** | **92.2** | **90.9** | **79.1** | **89.5** | **85.9** |
| | **GRASP** ($K$=32) | **0.004** | **90.4** | **93.8** | **83.4** | **61.9** | **91.5** | **90.3** | **77.2** | **88.2** | **84.6** |
| | Full-FT (Zaken et al., 2022) | 100 | 94.7 | 96.4 | 90.2 | 68.0 | 94.0 | 92.4 | 86.6 | 92.2 | 89.3 |
| | LoRA (Hu et al., 2021) | 0.30 | 94.9 | 96.2 | 90.6 | 68.2 | 93.4 | 92.6 | 87.4 | 91.6 | 89.4 |
| RoBERTa-Large | RED (Wu et al., 2024) | 0.02 | 93.5 | 96.0 | 89.5 | 68.1 | 90.3 | 91.3 | 86.2 | 88.9 | 87.9 |
| | VeRA (Kopiczko et al., 2023) | 0.02 | 94.4 | 96.1 | – | 68.0 | 90.9 | 91.7 | 85.9 | – | – |
| | **GRASP** ($K$=64) | **0.005** | **94.5** | **96.1** | **88.9** | **67.0** | **93.0** | **91.8** | **85.9** | **89.8** | **88.4** |

Table 1: GLUE benchmark comparison of GRASP-based fine-tuning on RoBERTa-Base and RoBERTa-Large against parameter-efficient fine-tuning baselines. %Param denotes the percentage of trainable parameters relative to the total model parameters.

Here, $\gamma, \beta \in \mathbb{R}^K$ are the learned scaling and shifting parameters for $K \ll D$ groups, where $D$ represents the input hidden dimension. These vectors are first repeated $m = \lceil D/K \rceil$ times and truncated to length $D$, yielding $\tilde{\gamma}, \tilde{\beta} \in \mathbb{R}^D$. A fixed permutation $\pi$ over $\{1, \ldots, D\}$ is then applied to obtain the permuted vectors $\hat{\gamma} = \tilde{\gamma}[\pi]$ and $\hat{\beta} = \tilde{\beta}[\pi]$. This effectively groups $D/K$ neurons to share a pair of learnable linear scaling and shifting parameters. For stable training, we initialize all $\gamma$ values to 1 and all $\beta$ values to 0.

## 3.2 Group Granularity

The number of groups ($K$) play a critical role in controlling the number of trainable parameters, as further explored in the Results section. Interestingly, reducing $K$ not only reduces trainable parameters but also encourages it to capture more salient features of the underlying data. We empirically validate this in Fig. 2b & c and Fig 3a. When parameters are learned independently (no grouping), their layer-wise distribution is approximately uni-modal Gaussian. As $K$ decreases, multiple-modes (mixture of Gaussians) arises in the parameter distribution, suggesting that the model is discovering and encoding distinct latent structures.

This effect becomes increasingly pronounced as $K$ is reduced from 128 to 8 (Fig. 3a), requiring the model to learn critical task-relevant features under tighter parameter constraints. In Fig. 3b, we demonstrate the trade-off between total trainable parameters and downstream task accuracy. Due to the observed non-linear relationship, we find that permitting a small drop in task-specific accuracy (i.e., $\leq 0.7\%$, from 94.3% to 93.6% on SST-2) enables a substantial additional reduction in trainable parameters—over $8\times$, achieved by decreasing $K = 128$ to $K = 16$.

## 4 Results

Following prior work (Zaken et al., 2021), we evaluate our proposed method on both masked and causal language modeling tasks. Specifically, we fine-tune RoBERTa-Base and RoBERTa-Large (Liu et al., 2019) on eight tasks of the GLUE benchmark (Wang et al., 2018) and GPT2-Medium (Radford et al., 2019) on the E2E NLG Challenge (Dušek et al., 2020), a data-to-text generation task. The experiments were done on Nvidia RTX A5000 GPUs (8) each with 24GB memory.

### 4.1 Natural Language Understanding Tasks

We present the results on GLUE benchmark using RoBERTa-Base and RoBERTa-Large models in Table 1. We report results on single sentence tasks such as CoLA and SST-2, on similarity-based tasks such as STS-B,

| Method | %Param | BLEU | NIST | METEOR | ROUGE-L | CIDEr |
|---|---|---|---|---|---|---|
| Full-FT[*] | 100 | 68.20 | 8.62 | 46.20 | 71.00 | 2.47 |
| FTTop2[*] | 7.10 | 68.10 | 8.59 | 46.00 | 70.80 | 2.41 |
| AdapterL[*] | 3.12 | 68.90 | 8.71 | 46.10 | 71.30 | 2.47 |
| LoRA[*] | 0.10 | 70.40 | 8.85 | 46.80 | 71.80 | 2.53 |
| Prefix Tuning[†] | 0.23 | 63.92 | 8.26 | 41.81 | 66.86 | 2.03 |
| $(IA)^{3†}$ | 0.05 | 63.63 | 7.99 | 40.49 | 66.36 | 1.89 |
| **GRASP** ($K$=384) | **0.020** | **68.64** | **8.76** | **45.00** | **69.47** | **2.37** |
| Adapter, rank 1[†] | 0.07 | 63.76 | 8.37 | 42.74 | 66.70 | 2.09 |
| Adapter-FFN, rank 1[†] | 0.02 | 62.99 | 8.09 | 40.88 | 66.39 | 1.98 |
| LoRA, rank 1[†] | 0.03 | 64.51 | 8.38 | 44.78 | 67.35 | 2.28 |
| RED[†] | 0.01 | 64.86 | 8.36 | 44.99 | 67.62 | 2.28 |
| **GRASP** ($K$=64) | **0.003** | **66.03** | **8.49** | **43.37** | **68.15** | **2.27** |

Table 2: Comparison of fine-tuning methods on the E2E NLG Challenge using GPT-2 Medium. %Param denotes the percentage of trainable parameters relative to full fine-tuning of GPT-2 Medium with 354.9M parameters. Results marked with [*] are from Hu et al. (2021); results marked with [†] are from Wu et al. (2024).

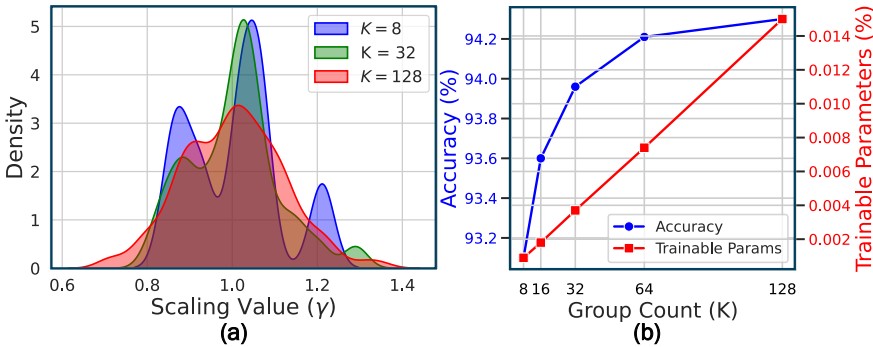

(a)  (b)

Figure 3: (a) KDE plots of scaling parameter ($\gamma$) distribution in the same projection layer (Key) for GRASP with varying group sizes $K \in \{8, 32, 128\}$. Smaller $K$ values produce distinct parameter clusters (modes), enabling the model to efficiently learn downstream task (GLUE SST-2) with significantly fewer trainable parameters. (b) Graph showing trade-off between accuracy and trainable parameters (%) on SST-2 task using GRASP with different values of $K$.

MRPC and QQP and natural language inference tasks such as RTE, QNLI, MNLI. We report matthews correlation for CoLA, pearson correlation for STS-B, F1-score for MRPC and accuracy for all other tasks.

For all the tasks in GLUE, we keep the maximum sequence length as 128. We use batch size of 16 for RTE, MRPC, MNLI and batch size of 32 for the rest of the datasets. The learning rate is set to $8 \times 10^{-4}$ for CoLA, RTE, and STS-B; $5 \times 10^{-4}$ for MNLI, QNLI, and QQP; and $1 \times 10^{-4}$ for SST-2 and MRPC. RoBERTa-Base has 12 layers with hidden dimension 768 and the intermediate dimension being 3072. RoBERTa-Large has 24 layers with a hidden dimension of 1024 and an intermediate dimension of 4096.

As highlighted in Table 1, our method surpasses prior approaches that rely on element-wise modulation, such as BitFit, $(IA)^3$, and RED. Remarkably, with only $K = 32$, GRASP attains comparable performance while reducing the number of trainable parameters by 75× **relative to LoRA**, 22.5× **relative to BitFit**, and 12.5× **relative to** $(IA)^3$.

## 4.2 Natural Language Generation Tasks

E2E datset comprises of 42K training, 4.6K validation and 4.6K testing samples. It is primarily used as data-to-text generation task.

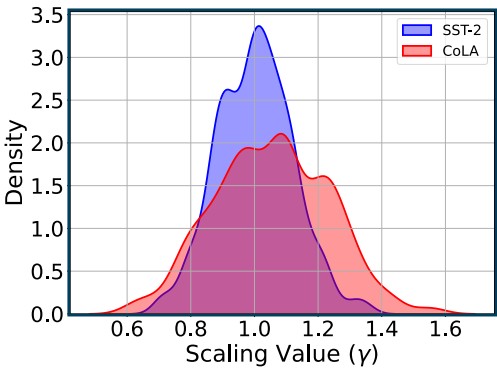

Figure 4: KDE plots of scaling parameter distribution of the same projection layer (Key) for GRASP on CoLA and SST-2.

For the E2E NLG Challenge, we use a train batch size of 4 and learning rate of $5 \times 10^{-4}$. We do beam-search during inference with a beam width of 10. The model dimension in GPT-2 Medium is 1024 and the intermediate dimension is 4096.

In the first part of Table 2, we compare GRASP with several popular PEFT strategies and demonstrate that it achieves competitive performance on generative tasks (E2E NLG Challenge), while using significantly fewer trainable parameters. To emphasize suitability of GRASP for low-resource environments, we compare it against low-rank LoRA and Adapters (with rank = 1), as well as the RED method in second part of Table 2. GRASP outperforms all of these approaches while requiring substantially fewer trainable parameters—up to **10× fewer than LoRA with rank = 1**.

## 4.3 Training Efficiency

We compare GRASP against LoRA ($r = 1$) on RoBERTa-large fine-tuned on SST-2 with a sequence length of 128 and batch size of 32. Despite its extremely low rank, LoRA introduces additional matrix multiplications during both the forward and backward passes, resulting in non-trivial computational overhead. In our experiments, LoRA required 10.90 GB of peak GPU memory and 10 min 45 s per training epoch. In contrast, GRASP performs lightweight additive parameter modulation without introducing additional matrix multiplications beyond those already present in the frozen backbone. Consequently, GRASP reduced peak GPU memory usage to approximately 9.1 GB and shortened training time to 9 min 30 s per epoch. Under FP32 training with the Adam optimizer (Kingma & Ba, 2014), each trainable parameter requires storage for the parameter itself and two optimizer moments. Beyond peak memory usage, GRASP also benefits from a substantially **smaller persistent training memory footprint**. Consequently, GRASP offers lower parameter and optimizer-state memory requirements while simultaneously reducing wall-clock training time. These results demonstrate that the parameter efficiency of GRASP translates into practical improvements in both computational efficiency and memory consumption during fine-tuning.

## 4.4 GRASP: Ablation Studies

We perform ablation studies to isolate the impact of different factors—namely, random grouping, the selection of layers for GRASP and the use of scaling versus shifting — on overall performance. The results are summarized next.

### 4.4.1 Random Grouping Robustness

Since we employ a random grouping strategy where random dimensions are grouped together, we do further exploration on what role different initial seeds play in model performance. We empirically report in Table 3 that the performance of the learnt model remains consistent across different starting configurations. This

| Method | QNLI | SST-2 | CoLA | STS-B |
|--------|------|-------|------|-------|
| GRASP ($K$=128) | $92.00 \pm 0.19$ | $94.27 \pm 0.11$ | $64.19 \pm 0.15$ | $90.89 \pm 0.03$ |

Table 3: Mean and standard deviation over three random grouping initializations for GRASP ($K = 128$) on selected GLUE tasks using RoBERTa-Base. The small variance across runs indicates that performance is largely insensitive to the specific grouping configuration.

| Target Layers | %Param | RTE | SST-2 | STS-B |
|---------------|--------|-----|-------|-------|
| All Linear Layers | **0.015** | **79.1** | **94.3** | **90.9** |
| $W_K$, $W_V$, FFN$_2$ | 0.008 | 78.3 | 94.1 | 90.4 |
| FFN$_2$ Only | 0.003 | 78.0 | 94.0 | 90.0 |

Table 4: Ablation study on RoBERTa-Base evaluating the effect of applying GRASP ($K = 128$) to different subsets of linear layers. Results indicate that adapting all linear layers provides the strongest performance while remaining highly parameter efficient.

validates that our proposed method enables dynamic adaption of learnt parameters based on the groups formalized without any degradation in performance.

### 4.4.2 Layer Selection

In Table 4, we conduct an experiment to compare the performance of GRASP when applied in three different ways: (1) to all linear sub-layers of an encoder block—similar to BitFit, but excluding LayerNorm layers; (2) selectively to specific components such as the key ($K$), value ($V$), and second feedforward projection ($FF2$) layers, as in $(IA)^3$; and (3) only to the final $FF2$ layer. By only limiting GRASP to FF2 layer, we achieve competitive performance while reducing total trainable parameters significantly.

### 4.4.3 Effect of Scaling & Shifting

In Table 5, we isolate the effect of learning scaling and shifting parameters. We find learning only the shift parameter results in improved performance compared to learning only scaling.

### 4.4.4 Effect of Tasks on Parameter Distribution

We further analyze how the per-layer parameter distributions vary across tasks of different difficulty levels. As shown in Fig. 4 comparatively harder task such as CoLA induces a parameter distribution with more pronounced multi-modality than an easier task like SST-2, even when using the same group size ($K = 128$). This suggests that GRASP adapts its learned perturbations to the underlying task complexity, effectively capturing task-critical structure despite operating with a limited number of trainable parameters.

## 5 Stochastic GRASP

The multi-modal structure of the learned shift parameters, evident in Fig. 3a and increasingly pronounced at smaller $K$, motivates our stochastic formulation, StochGRASP. Rather than treating this multimodality as an artifact of aggressive grouping, we interpret it as evidence that fine-tuning perturbs the pre-trained weights of each modulated layer according to an underlying distribution. We therefore reformulate PEFT as the task of learning, per layer, a set of $K$ Gaussian perturbations parameterized by their means ($\mu$) and standard deviations ($\sigma$), rather than fixed deterministic updates. Beyond providing a principled account of the observed multimodality, this distributional view suggests a path to robustness: by modeling perturbations explicitly during training, the model may internalize variability of the kind encountered at inference time on noisy or non-ideal hardware—a hypothesis we examine in Section 5.2.

| Method | RTE | SST-2 | STS-B |
|--------|-----|-------|-------|
| Without Shifting | 77.9 | 93.0 | 90.1 |
| Without Scaling | 78.7 | 93.8 | 90.4 |
| Full GRASP | **79.1** | **94.3** | **90.9** |

Table 5: Ablation study of the scaling and shifting components in GRASP ($K = 128$) on selected GLUE tasks using RoBERTa-Base. Removing either component degrades performance, indicating that both contribute to the effectiveness of the method.

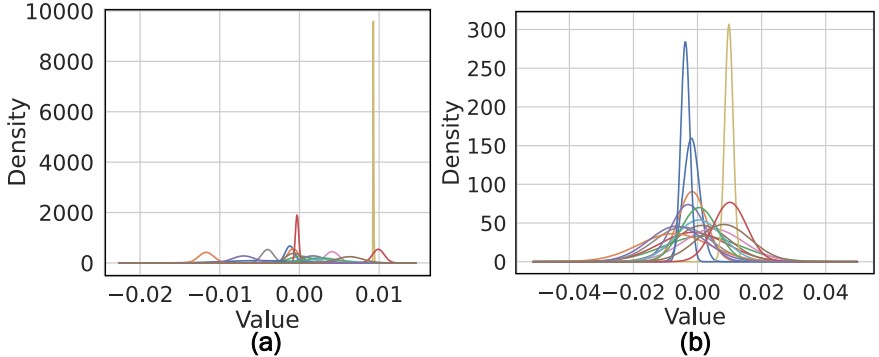

(a)  (b)

Figure 5: (a) Gaussian distributions (perturbations) learnt without the modified objective, and (b) distributions learned with the proposed objective (Eqn. 6); both learns $K = 16$ distributions per layer.

Our stochastic formulation, relates GRASP's activation-space shifts to an equivalent structured perturbation in weight parameter space. Consider a frozen linear sub-layer

$$Y = XW + b, \qquad W \in \mathbb{R}^{D_{\text{in}} \times D_{\text{out}}}, \ b \in \mathbb{R}^{D_{\text{out}}}. \tag{3}$$

In the simplified (shift-only) GRASP setting, the input activations are modified group-wise before the linear map. Let each input dimension $j$ belong to a group $g(j) \in \{1, \dots, K\}$, with a learned shift $t_{g(j)}$. Defining the shift vector $t \in \mathbb{R}^{D_{\text{in}}}$ via

$$\tilde{x}_j = x_j + t_{g(j)} \quad \implies \quad \tilde{X} = X + T,$$

the transformed output becomes

$$\tilde{Y} = \tilde{X}W + b = (X + T)W + b = XW + \underbrace{(TW + b)}_{b_{\text{eff}}}. \tag{4}$$

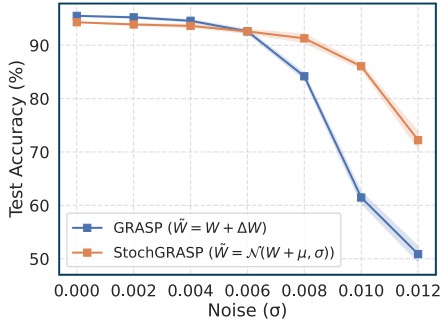

Figure 6: Robustness comparison between StochGRASP and a deterministic variant of GRASP on the SST-2 evaluation set using RoBERTa-Large. We inject varying levels of hardware-level noise during inference and report performance averaged over three random seeds, each corresponding to a distinct noise realization.

Thus, applying group-wise shifts in the activation space is *functionally equivalent* to introducing an effective bias term $b_{\text{eff}} = b + TW$, which is broadcast across all input rows, while $W$ and $b$ remain frozen.

## 5.1 Formulation

Empirically, as the group size decreases, the learned GRASP shifts $\{t_{g(j)}\}_{j=1}^{D_{in}}$ exhibit clear multi-modal structure across layers. Since the mapping $T \mapsto TW$ in Eqn. 4 is linear, multi-modal distribution over the group-wise shifts $T$ can induce a corresponding multi-modal distribution over the effective bias $b_{\text{eff}}$. This suggests reinterpreting fine-tuning as the injection of *distributed* perturbations rather than fixed deterministic offsets. We adopt this distributional view and, generalizing from the bias to the full weight matrix, formulate StochGRASP as learning Gaussian perturbations over the pre-trained weights (Eqn. 5), of which the bias-only case is a special instance.

While Eqn. 1 applies deterministic scaling and shifting to activations, our proposed StochGRASP instead models the trainable update as a distribution over weight perturbations. We assume the learnt perturbation to follow underlying *Gaussian Distributions*. After training, the effective weights can be written as:

$$
\begin{aligned}
\tilde{W}_{i,j} &= W_{i,j} + \Delta W_{i,j}, \qquad i \in \{1, \ldots, D_{\text{in}}\},\ j \in \{1, \ldots, D_{\text{out}}\}, \\
\Delta W_{i,j} &\sim \mathcal{N}\big(\mu_{f(i),\,g(j)},\ \sigma_{f(i),\,g(j)}\big), \\
\tilde{W}_{i,j} &\sim \mathcal{N}\big(W_{i,j} + \mu_{f(i),\,g(j)},\ \sigma_{f(i),\,g(j)}\big).
\end{aligned}
\tag{5}
$$

where $g(j) \in \{1, ..., K\}$ and $f(i)$ controls how many distinct Gaussian distributions are learned for each row of weight matrix: Setting $f(i) = i$ learns separate set of distributions for each input row, yielding $O(D_{in} * K)$ trainable parameters. Setting $f(i) = 1$ shares a single distribution across all rows for a specific column, resulting in $O(K)$ trainable parameters.

**Thus, rather than learning deterministic weight updates, we formulate PEFT as the stochastic task of learning Gaussian distributions whose samples generate perturbations to align (fine-tune) the pre-trained model with a given dataset.** This not only provides a principled explanation for the emergent multi-modal behavior but also yields a model that is inherently more resilient to noise, particularly in edge-level or compute-constrained hardware settings.

## 5.2 Robustness of StochGRASP

As discussed in Section 5, StochGRASP reformulates PEFT as the task of learning a probability distribution over the weight perturbations required for downstream adaptation. To further align the learned perturbations with hardware-level noise characteristics (e.g., variability in analog non-volatile memory devices), **we introduce an additional regularization term that encourages the learned standard deviations to match a hardware-motivated target value**. The modified training objective becomes:

$$
\mathcal{L}_{\text{total}} = \mathcal{L}_{\text{task}} + \frac{\lambda}{L} \sum_{\ell=1}^{L} \sum_{i=1}^{D_{\text{in}}^{(\ell)}} \sum_{j=1}^{D_{\text{out}}^{(\ell)}} \left( \sigma_{f(i),g(j)}^{(\ell)} - \sigma_{\text{target}} \right)^2
\tag{6}
$$

where $\mathcal{L}_{\text{task}}$ denotes the original fine-tuning loss, $\sigma^{(\ell)}$ represents the learned standard deviation vector for layer $\ell$, $\sigma_{\text{target}}$ is the desired hardware-aligned noise level, and $\lambda \in [0.01, 0.1]$ controls the strength of the regularization. During training, we initialize the mean of the learned Gaussian perturbations as 0, and set the standard deviation as $\sigma \in [0.001, 0.005]$, resulting in a stable initialization. Fig. 5b shows the learned parameter distributions after applying the proposed loss function (with $\sigma_{target} = 0.05$). Compared to Fig. 5a (without the modified objective), we observe a clear increase in the learned standard deviations ($\sigma$), indicating that the objective explicitly encourages broader distributions that provide greater robustness to parameter variability (noise) during inference.

### 5.2.1 Why Gaussian Perturbations?

Gaussian perturbations are widely adopted as a first-order approximation for stochastic non-idealities in analog compute-in-memory (CiM) and neuromorphic accelerators, particularly when multiple independent error sources jointly contribute to effective computation noise. Prior studies (Büchel et al., 2026; Gokmen & Vlasov, 2016; Manna et al., 2024) have modeled dominant hardware effects—including thermal noise, ADC quantization noise, read noise, and cycle-to-cycle device variability—using additive Gaussian perturbations during training and inference to capture aggregate uncertainty in analog execution. Motivated by this convention, for preliminary investigation, we evaluate StochGRASP under additive Gaussian weight perturbations, which serve as a hardware-agnostic proxy for accumulated analog inference noise. While this does not capture all device-specific effects such as drift, retention loss, or asymmetric conductance updates, it provides a standard and controlled setting for assessing whether stochastic fine-tuning improves tolerance to inference-time variability.

### 5.2.2 Results

In Fig. 6, we assess the robustness of StochGRASP under varying levels of simulated noise and compare it against a deterministic variant of the GRASP baseline. For this experiment, we train two separate models: (i) a deterministic GRASP model, that learns $K = 64$ fixed shift parameters per row of each modulated linear layer, effectively training without any stochasticity, and (ii) our StochGRASP variant, which instead learns $K$ Gaussian distributions per row of each modulated layer. To ensure a fair comparison, the deterministic GRASP baseline learns shift parameters directly in the weight space rather than the activation space, matching the parameterization employed by StochGRASP. This isolates the effect of stochasticity and uncertainty modeling from differences in the underlying adaptation mechanism. During inference, both models are evaluated under varying levels of injected Gaussian noise. For StochGRASP, instead of sampling from the learned standard deviation, we replace it with the standard deviation corresponding to the injected noise level. This situation is equivalent to deploying the learnt mean values of the fine-tuned weight parameters to an edge AI hardware accelerator that is associated with an independent noise for each programmed weight. Each setting is tested over three random seeds to capture variability in noise realizations. Since, StochGRASP learns *distributions* rather than fixed values, it models weight perturbations during training and internalizes this variability. **Specifically, under normal noise ($\sigma = 0.01$), StochGRASP maintains accuracy of $\geq 86\%$ while the deterministic variants deteriorates to $61.5\%$ on SST-2 dataset using RoBERTa-Large model.** As a result, StochGRASP demonstrates markedly higher tolerance to hardware-level noise, yielding significantly better performance across all disturbance levels.

## 6 Conclusions

We introduced GRASP, a lightweight PEFT method that uses group-wise activation modulation to drastically reduce trainable parameters while matching or exceeding established PEFT baselines on language understanding and remaining competitive on generation. We showed that aggressive compression (small $K$) induces multimodal parameter structure, and that this structure sharpens with task difficulty—evidence that GRASP encodes task-relevant information under tight parameter budgets. Interpreting this multimodality as a signature of distributed weight perturbations, we proposed StochGRASP, which learns per-group Gaussian perturbations together with a noise-aware objective. In preliminary experiments under simulated inference-time noise, StochGRASP shows substantially better tolerance than a deterministic baseline, suggesting that distributional PEFT is a promising route to robust inference on noise-prone edge and analog AI hardware. Validating this on hardware-calibrated noise models—and ultimately on real edge accelerators—is an important direction for future work.

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
