# OpenReview forum: "GRASP: Grouped Activation Shared Parameterization for Parameter-Efficient Fine-Tuning and Robust Inference of Transformers"
_TMLR — Under review for TMLR_

### Review · Reviewer_d87V · 2026-07-04

**Summary Of Contributions:**

This paper introduces GRASP, a parameter-efficient fine-tuning method. Its core approach involves randomly partitioning the input activation dimensions of specific linear layers within a Transformer into multiple groups, with each group sharing a set of scaling and shifting parameters. Experiments on the GLUE and E2E NLG benchmarks demonstrate that GRASP achieves competitive performance while utilizing a minimal number of trainable parameters.

The paper also introduces StochGRASP. Observing that the distribution of learned parameters exhibits a multimodal structure when the number of groups is small, the authors reframe fine-tuning as learning a distribution of weight perturbations. StochGRASP learns parameters for Gaussian perturbations and demonstrates better robustness compared to a deterministic GRASP variant under simulated inference noise.

The strengths of this work lie in its simplicity, minimal parameter count, and seemingly low implementation cost; as an activation modulation method requiring very few parameters, GRASP may hold practical value.

**Additional Comments:**

Overall, I believe the problem addressed in this paper is promising; however, it requires fairer baselines, a clearer narrative thread, and more robust experimental support for StochGRASP.

**Audience:**

Yes

**Audience Explanation:**

I think a segment of TMLR readers will likely be interested in this paper. Parameter-efficient fine-tuning is a significant area of ​​research, particularly for scenarios involving large model deployment, low-resource fine-tuning, and storage constraints.

**Broader Impact Concerns:**

I do not see any significant ethical risks that require special emphasis.

**Claims And Evidence:**

No

**Claims Explanation:**

I believe some of the claims in the paper are supported by evidence, but the current evidence is insufficient to fully substantiate the authors' stronger conclusions.

Regarding GRASP, the experiments do demonstrate that it achieves solid GLUE and E2E results using very few trainable parameters. However, the current comparisons lack proper control of variables. The authors primarily compare GRASP against methods like LoRA, Adapter, BitFit, IA3, and RED using their default settings. While this comparison shows that GRASP uses fewer parameters without a catastrophic drop in performance, it does not adequately demonstrate that GRASP is superior to these methods in terms of parameter efficiency. A fairer experimental setup would compare the performance of different methods under the same parameter budget, or compare the number of parameters required by each method to achieve a specific performance level.

Another issue is the absence of a "no fine-tuning" baseline. If the authors' main claim is that "competitive performance can be achieved with minimal parameters," it is essential to know the performance level without any fine-tuning. Otherwise, it is difficult for readers to determine whether the gains from GRASP stem from effective fine-tuning or simply from the model's inherent strength on these tasks.

The explanation regarding multimodality also lacks sufficient evidence. The observation that the learned parameter distribution becomes multimodal as the number of groups decreases is interesting, but it does not necessarily imply that the model has learned the latent task structure; it could simply be a natural consequence of parameter sharing and compression. The authors need more direct evidence to support this interpretation.

The evidence for StochGRASP is even weaker. StochGRASP is compared only against a deterministic GRASP variant under simulated Gaussian noise. While this experiment shows that stochastic training benefits this specific baseline, it does not demonstrate that StochGRASP outperforms existing PEFT methods. The authors should include comparisons with methods like LoRA, IA3, and RED under the same noise settings, or include noise-aware versions of relevant baselines.

**Requested Changes:**

**1. Include a baseline comparison based on an equivalent parameter budget.**

Currently, Tables 1 and 2 primarily demonstrate that GRASP achieves strong performance despite using fewer parameters, but the parameter count is not adequately controlled. The authors should compare the performance of GRASP, LoRA, Adapter, IA3, RED, and other methods using the same trainable parameter ratio. Alternatively, they could compare the number of trainable parameters required by each method to achieve the same level of performance. This would provide clearer support for the conclusion that GRASP offers superior parameter efficiency.

**2. Include a "no fine-tuning" baseline.**

The authors should report results on GLUE and E2E using the frozen model without any fine-tuning. This would allow readers to assess the actual performance gain provided by GRASP compared to a scenario with no fine-tuning at all. This is a crucial reference point for a paper that emphasizes fine-tuning with minimal parameters.

**3. Clarify the relationship between GRASP and StochGRASP.**

The first half of the paper introduces GRASP (operating in activation space), while the second half introduces StochGRASP (operating in weight space). The authors need to clarify the following points: What is the ultimately recommended method? Should GRASP or StochGRASP be used for standard PEFT scenarios? Are the two methods comparable in terms of parameter count, training cost, and task performance?

**4. Standardize the use of bold formatting in Table 1 and Table 2.**

Currently, the bolded results in the tables do not always represent the best performance for each column; instead, they highlight the authors' own method. This may confuse readers. It is recommended to either bold the best result in each column—following standard conventions—or clearly define the meaning of the bold formatting.

**5. Provide additional experimental details.**

For instance, clarify whether hyperparameters were re-tuned for the different baselines, and whether the same training epochs, batch sizes, learning rate search ranges, and model checkpoints were used. These details are crucial for PEFT comparisons.

---

### Review · Reviewer_cy9q · 2026-07-08

**Summary Of Contributions:**

## Summary

The authors introduce Grouped activation shared parameterization (GRASP), a parameter-efficient fine-tuning (PEFT) framework for pre-trained transformers. Instead of optimizing all parameters separately, GRASP learns shared linear scale and shift parameters for parameters that are grouped within each layer. By randomly partitioning the neurons in K groups, where K is much smaller than the total layer dimension, GRASP drastically reduce the number of trainable parameters considered in the fine-tuning process. At the same time, experiments on the GLUE benchmark and E2E NLG language challenge show that GRASP achieves competitive results, nearing or outperforming the performance of the baselines (including LoRA, Adapters, BitFig and $(IA)^3$.

Whilst the number of groups (K) play a critical role in regulating the number of trainable parameters, reducing K seems to encourage the algorithm to capture more salient features. This observation is reflected in the distribution of learned shift and scale parameters, that goes from a uni-modal Gaussian distribution for large values of K to a predominantly more multimodal distribution when K decreases. This multimodal structure motivates the introduction of StochGRASP, a stochastic variant of GRASP that is more robust against hardware-induced noise.

---

I thank the authors for their interesting proposal. I first summarize the strengths of the proposal, as well as my questions and concerns below.

### Strengths
* **Relevance**: Currently, pre-trained transformers are ubiquitous, and a lot of efforts have been made to efficiently adapt the general models to domain-specific tasks. With the introduction of GRASP, the authors contribute towards this goal. Albeit the model does not always outperform the state-of-the art fine-tuning methods, GRASP excels in scenarios with limited computational resources. Therefore, the method is expected to be of interest to plenty researchers.
* **Coverage**: The authors introduce a new framework for fine-tuning pretrained transformers. The framework is extended with a nondeterministic variant.
* **Baselines & Benchmarks**: The authors demonstrate the effectiveness of their algorithm on the GLUE benchmark (Wang et al., 2018) and the E2E NLG Challenge (Drušek et al., 2020). The algorithm is tested against a wide range of sensible baselines, including LoRA, $(IA)^3, and AdapterL.
* **Ablation studies**: the work features many ablation studies, where including an interesting selection assessment (Section 4.4.2) which ties the current work to the layer-selective related work FFTTop2 (Hu et al., 2021).

### Weaknesses and questions

**Weaknesses**
* **Highlighting Figure 1**: Figure 1 provides a solid overview of the presented work, but it is not highlighted in the text (or I missed it, sorry). Because of that, some components, like $\beta_g$ are only introduced a lot later, which jeopardizes the figure a little. I can recommend making the figure either more high-level, or introducing all terms presented in the figure earlier on.
* **Related work**:  The related work section could be more elaborated. Currently, the authors present the related work as is and omit a comparison with their own method. Therefore, it is currently unclear what the “critical gap in existing approaches” (page 3, bold) entails.
* **Results citations**: Section 4.1 introduces many natural language tasks without elaborating on their content. The section could therefore benefit from adding the citation for each natural language task or briefly explaining the goal of each task in the appendix.
* **Results highlighting**: In table 1 and 2, all results obtained with GRASP are highlighted in bold, even though these results are not guaranteed to be the best. This is slightly misleading, and I would therefore recommend to either highlight the best-performing model, or not highlight any results at all.
* **Random grouping robustness**:  Whilst the ablation study on random grouping robustness is very interesting, it is currently too limited. I would expect that the grouping indifference to permutations depends on the group size K. However, this aspect is currently unexplored, as the authors only consider K=128 in Table 3. I think that the manuscript would benefit from a larger ablation on this matter. More specifically, would it be an idea to “absorb” this ablation in the main results presented in Table 1 and 2? In essence, the randomness of the groups is a part of the algorithm that introduces some uncertainty in the performance. Reporting this uncertainty with the main results greatly improves the transparency thereof.
* **Model choice**: RoBERTa and GPT-2 are relatively old models. Since then, transformer architectures underwent a lot of changes, increasing their attention efficiency, position tracking and layer routing. If feasible, I would specifically be very interested how GRASP performs with Mixtures of Experts (MoE). I think that the inclusion of one of the state-of-the-art transformers in the evaluation would greatly improve the relevance of the paper.

**Minor comments**
* **Spelling and grammar**: The manuscript is in a good shape, but there are some minor grammatical errors such as “The number of groups (K) play” (Section 3.2)

**Questions**

* **Stable training**: The authors mention that “For stable training, we initialize all $\gamma$ values to 1 and all $\beta$ values to 0”. Could the authors elaborate on why this induces stability in the training?

**Audience:**

Yes

**Audience Explanation:**

The increased prevalence and use of (large) pre-trained transformers have led to a surge of interest fine-tuning methods that can be applied with limited resources. While notable techniques such as LoRA and Adapters achieve a strong performance, this work introduces an even more parameter efficient fine-tuning algorithm. Therefore, I think that the introduction of GRASP, and the non-deterministic StochGRASP variant, will be of interest to TMLR’s audience.

**Broader Impact Concerns:**

There seem to be no ethical implications of the work that go beyond the implications associated with generative models, such as large language models. Therefore, I consider adding a broader impact statement optional.

**Claims And Evidence:**

Yes

**Claims Explanation:**

The main claims regarding the performance of the introduced framework are sufficiently supported by experimental evidence.

**Requested Changes:**

The requested changes are based on the weaknesses identified above.

* Add a textual description of Figure 1.
* Elaborate the Related Work, highlighting on the difference between the related and current works.
* Elaborate on the natural language tasks from Section 4.1.
* Absorb the ablation study on the random grouping robustness into the main results (Table 1 and 2).
* If feasible, apply the algorithm to a more recent transformer architecture to enhance the relevance of this work.

---

### Review · Reviewer_pkWD · 2026-07-19

**Summary Of Contributions:**

The paper proposes GRASP, which reduces PEFT parameters by sharing activation scaling and shifting values across randomly grouped hidden dimensions. It further introduces StochGRASP, a stochastic weight-perturbation variant intended to improve robustness to hardware noise. The method is simple and achieves promising results at very small parameter budgets. However, the key design choices are not sufficiently validated, and the evidence for the stochastic extension remains preliminary.

**Audience:**

Yes

**Audience Explanation:**

The ultra-low-parameter PEFT setting is relevant to researchers interested in efficient adaptation, and robustness to noisy inference hardware is a worthwhile direction. The current results are preliminary, but the underlying questions are of interest to many of the TMLR audience.

**Claims And Evidence:**

No

**Claims Explanation:**

The experiments support the narrow claim that grouped scale-and-shift adaptation can work well at very small parameter budgets on the reported settings. They do not fully support the broader claims about the design rationale, scalability, robustness, or practical efficiency.

In particular, random grouping is tested only through altering random seeds, rather than against other equally cheap grouping rules. The layer-selection and scale/shift ablations also change the number of trainable parameters, making their conclusions difficult to isolate. The effect of K is shown only on SST-2, so it is unclear whether GRASP provides a stable accuracy–parameter trade-off across tasks.

The evaluation of StochGRASP is not adequate as the method is evaluated on one model and one task, without a deterministic baseline trained with the same noise injection. Moreover, the learned standard deviation is replaced by the test noise level during inference, which makes the role of the learned distribution unclear. The efficiency result is also based on a single RoBERTa/SST-2 comparison.

**Requested Changes:**

1. The empirical evaluation is currently too narrow for the paper’s general PEFT claims. RoBERTa/GLUE and GPT-2/E2E may remain as controlled reference settings, but the paper should also evaluate a modern decoder-only, instruction-tuned model on several task types. A more informative extension would be to evaluate a current open-weight LLM family on both reasoning and instruction-following or generation tasks, with the main PEFT baselines compared under matched parameter and tuning budgets.

2. The experiments do not yet establish that the proposed design choices are preferable to simpler alternatives. Random grouping should be compared with contiguous grouping and at least one simple structured grouping at the same parameter budget. The layer-placement and scale-versus-shift ablations should likewise control for parameter count and be repeated across more than one task. The paper should also position the contribution more precisely: GRASP appears to add parameter sharing to an existing scale-and-shift adaptation mechanism, while the effective-bias derivation does not by itself justify the full weight-space formulation used by StochGRASP.

3. The useful operating range of GRASP should be made clear. The current $K$ analysis is limited to SST-2 and mainly demonstrates an extreme low-parameter regime. Please report accuracy–parameter curves across several tasks, including multiple $K$ values, the ungrouped $K=D$ case, matched-budget PEFT baselines, and full fine-tuning as a reference. Because the $K=D$ limit is still a per-dimension affine transformation rather than an unrestricted weight update, the paper should either demonstrate a reasonably broad effective range or explicitly position GRASP as an ultra-low-budget method.

4. The robustness experiment does not isolate the benefit of the stochastic parameterization. StochGRASP should be compared with a deterministic model trained using exactly the same noise-injection schedule. The paper should also include an ablation that retains the learned standard deviations at inference and explain the rationale for replacing them with the externally imposed test-noise level in the main experiment. Evaluation on another task or model, together with basic ablations over $f(i)$, $\lambda$, and $\sigma_{\text{target}}$, is needed before drawing broader conclusions about robustness to hardware noise.

5. The practical costs and potential deployment benefits require stronger evidence. Please report the exact parameter count, optimizer-state cost, memory footprint, and training throughput of both GRASP and StochGRASP in at least one modern decoder-only setting. The paper should also clarify what (if any) additional implementation cost arises for commonly used quantized or fused linear layers.